# Engineering the Aggregation of Dyes on Ligand-Shell Protected Gold Nanoparticles to Promote Plexcitons Formation

**DOI:** 10.3390/nano12071180

**Published:** 2022-04-01

**Authors:** Nicola Peruffo, Giovanni Parolin, Elisabetta Collini, Stefano Corni, Fabrizio Mancin

**Affiliations:** 1Department of Chemical Sciences, University of Padova, Via Marzolo 1, 35131 Padova, Italy; nicola.peruffo@unipd.it (N.P.); giovanni.parolin@studenti.unipd.it (G.P.); 2Padua Quantum Technologies Research Center, Via Gradenigo 6, 35131 Padova, Italy

**Keywords:** plexcitons, polaritonic chemistry, gold nanoparticles, porphyrins, J-aggregates, templating

## Abstract

The ability to control the light–matter interaction in nanosystems is a major challenge in the field of innovative photonics applications. In this framework, plexcitons are promising hybrid light–matter states arising from the strong coupling between plasmonic and excitonic materials. However, strategies to precisely control the formation of plexcitons and to modulate the coupling between the plasmonic and molecular moieties are still poorly explored. In this work, the attention is focused on suspensions of hybrid nanosystems prepared by coupling cationic gold nanoparticles to tetraphenyl porphyrins in different aggregation states. The role of crucial parameters such as the dimension of nanoparticles, the pH of the solution, and the ratio between the nanoparticles and dye concentration was systematically investigated. A variety of structures and coupling regimes were obtained. The rationalization of the results allowed for the suggestion of important guidelines towards the control of plexcitonic systems.

## 1. Introduction

The porphyrin dye 5,10,15,20-tetrakis−4-sulfonato-phenyl porphyrin (TPPS) has been intensely studied over the years [1,2,3]. This molecule is indeed quite unique, since it conjugates the relevant photophysical properties of porphyrins with a strong tendency to form aggregates with peculiar optical and structural features [4,5,6,7,8,9,10,11,12].

At a basic or neutral pH, TPPS bears four negative charges located at the peripheral sulfonate groups and no charge at the inner pyrrolic nitrogens (TPPS^4−^, Figure 1a). The sulfonate groups provide the solvation necessary to grant solubility in water and prevent aggregation by electrostatic repulsion. When the pH is reduced, the protonation of the two inner pyrrolic nitrogens (pK_a_ ~ 4.8) takes place (H_2_TPPS^2−^, Figure 1a) [13]. The different charge distribution of H_2_TPPS^2−^ results in the tendency to form aggregates of the J-type (parallel displaced), due to the reduced repulsion, the better surface overlap, and the possibility of charge pairing [10,13,14,15,16]. On the other hand, the formation of aggregates of the H-type (sandwich) remains hampered because of the unfavorable charge distribution [17].

Notwithstanding the smaller electrostatic repulsion caused by the decrease of the pH, the formation of aggregates is still an unfavorable process and requires other factors, such as the increase of the dye concentration, the increase of the ionic strength of the medium and, most relevant, the presence of templating agents [15,18,19,20]. Indeed, polycationic species, such as cationic polymers or peptides, usually induce the formation of J-aggregates, even at low concentrations and ionic strength [17].

Cationic nanoparticles (NPs) should have a similar templating capability, but, unlike the previous examples, they provide a 3D organization of the positive charges, potentially leading to different self-assembly arrangements of the dyes. Indeed, the few examples reported so far reveal that distinct structures are obtained, depending on the properties of the NPs and the conditions used. Gold nanorods [14,21] and carbon nanodots [22] coated with positively charged species were shown to template the formation of J-aggregates of H_2_TPPS^2−^ at an acidic pH and the formation of H-aggregates of TPPS^4−^ at a basic pH. Cationic maghemite (γ-Fe_2_O_3_) nanoparticles, on the other hand, did not template any aggregation. Instead, they induced the partial deprotonation of the H_2_TPPS^2−^ dye molecules absorbed on the particle surface to get the TPPS^4−^ form, even at an acidic pH [23]. These apparently contradictory examples confirm that different and, so far, partly elusive parameters control the self-assembly of the porphyrin on cationic NPs.

Remarkably, the assembly of dyes on plasmonic nanostructures, including NPs, can lead to the formation of hybrid systems allowing the control and exploitation of light–matter interaction at the nanoscopic scale [24,25,26]. In particular, when a strong coupling is established among the plasmonic and the excitonic moieties, hybrid polariton states, called plexcitons, may form [27,28]. The formation of plexcitons has been extensively studied with metal surfaces endowed with propagating surface plasmon polaritons (SPPs) [29,30], and it was recently extended to the localized surface plasmon resonances (LSPRs) of metal NPs in solution [31,32,33,34]. Interest in plexcitons is justified by the possibility of controlling the matter properties just by acting on the light–matter coupling, enhancing the efficiency of relevant reactions such as energy and electron transfer [35,36], and reducing the interactions with the environment [37]. These features are promising for important applications in artificial light-harvesting, sensors, and photonics [25,26,28,36,38]. Plexciton materials, where the plasmonic component is a colloidal nanoparticle, are rising increasing interest because they are tunable, scalable, and easy to synthesize by cheap wet chemistry methodologies [39,40].

Most of the plexcitons with NPs so far reported have been obtained using dye aggregates as the molecular excitonic component, since the interaction with the plasmon is reinforced by the high transition dipole moment promoted by aggregation. Consequently, the formation of plexcitons between NPs and individual dyes in solution has been observed in a limited number of cases [41,42,43,44].

Recently, we reported a multi-plexciton system formed by assembling H_2_TPPS^2−^ on 11 nm gold NPs coated with the cationic thiol *N*,*N*,*N*-trimethylammonium octane thiol (TMAO-SH, Figure 1b) [24]. In this system, two sets of plexciton resonances appeared, resulting from the coupling of the nanoparticles’ plasmon with electronic states of H_2_TPPS^2−^, both in the J-aggregate and monomeric forms. The observation of plexciton resonances arising from the coupling of the Q band of the monomeric porphyrins with the particles was particularly intriguing. Indeed, several factors were potentially disfavoring its formation: the extinction coefficient of the Q band of the H_2_TPPS^2−^ monomer was modest, its detuning [45] with the NPs plasmon was relatively high, and the most likely conformation of monomeric dyes on the particle surface was not granting the correct alignment of the transition dipoles (see infra). We suggested that this peculiar behaviour could be attributed to two concurring and related phenomena: (i) the aggregation of the NPs induced by H_2_TPPS^2−^ that red-shifts the particles’ plasmon to a more favorable wavelength; (ii) the consequent formation of plasmon nanogaps [46] entrapping the H_2_TPPS^2−^ molecules, where the strongly confined electromagnetic field enhances the dyes’ absorption. However, the reasons for the formation of such peculiar structures remained unclear.

We hence decided to perform a comprehensive investigation of the effects of different parameters, i.e., size of the NPs, pH, and concentration, in the interaction between NPs and porphyrins. The experimental studies have been complemented by classical atomistic molecular dynamics (MD) simulations to get insights into the most favorable supramolecular arrangements. The results obtained suggest that the peculiar interaction of the dye with charged monolayers, the NPs size, the concentration, and the pH are the key parameters that control the fate of the system, disclosing the possibility to exploit nanoparticle templates to engineer the physicochemical properties of the hybrid assemblies.

## 2. Materials and Methods

### 2.1. Big Nanoparticles (BNPs) Synthesis

BNPs were prepared as previously described [24]. Citrate-capped NPs were prepared following a modified Turkevich protocol [47]. The citrate capping layer was subsequently replaced with *N*,*N*,*N*-trimethylammonium octane thiol (TMAO-SH) to make the NPs’ surface positively charged [48]. A transmission electron microscopy (TEM) analysis (recorded on a Jeol 300 PX electron microscope, JEOL, Tokyo, Japan) revealed an average diameter of 11 ± 2 nm. From these data and the results of a thermogravimetric analysis (TGA) performed with a TA Q5000 IR instrument (TA Instruments, New Castle, DE, USA) we obtained an averaged formula of Au_30891_(TMAO-SH)_2426_. According to this formula, for convenience, the nanoparticle concentration has been converted into a concentration of 8-trimethylammonium octylthiol units grafted on their surface by multiplying by 2426. A nuclear magnetic resonance (NMR) analysis, performed with a Bruker AV III 500 spectrometer (Bruker, Billerica, MA, USA), confirmed the purity of the sample (Appendix A).

### 2.2. Small Nanoparticles (SNPs) Synthesis

SNPs were prepared following a modified Brust and Scriffin protocol [49]. The native dioctylamine capping agent was exchanged with TMAO-SH. The TEM analysis revealed an average diameter of 2.6 ± 0.7 nm (Appendix A). From these data and the results of the TGA (Appendix A), we obtained an averaged formula of the nanoparticles that was Au_566_ (TMAO-SH)_241_. An NMR analysis confirmed the purity of the sample (Appendix A). The SNPs concentration was converted into TMAO-SH units grafted on their surface by multiplying by 241.

Additional details about the synthesis and characterization of the thiol and the NPs are described in the Appendix A.

### 2.3. Synthesis of The Nanosystems

NPs–porphyrin hybrid samples in acidic or basic conditions were prepared by adding, in the following order, appropriate volumes (Table 1) of HCl solution at pH = 2.2 or NaOH solution at pH = 11 and 5 mM NPs solution to a 1 mM solution of H_2_TPPS^2−^ at pH 3.2 (or to a 1 mM solution of TPPS^4−^ at pH 11). Volumes were adjusted to obtain samples with a different Particle Area Per Porphyrin (PAP) parameter, which expresses the dye concentration in terms of the ratio between the number of porphyrin molecules in the sample and the average nanoparticle surface area (nm^2^). We omitted the part of the acronym regarding the pH because the volumes do not change as a function of it.

After the preparation, the samples were incubated for 8 h at room temperature, and then, their optical properties were measured. Still, a time-dependent analysis demonstrated that the samples were stable over this time range and that their optical properties did not change significantly.

### 2.4. Computational Methods

All MD simulations and analyses were performed with the GROMACS suite of packages [50]. The OPLS-AA [51] parameters for the H_2_TPPS^2−^ molecule were obtained with the LigParGen web server [52]. The structure and topology for the Au_144_L_60_ (L = S(CH_2_)_8_NH_3_^+^ = S-AO) cluster were derived from previous works [53], and the Au core was kept frozen during the dynamics. The Au (111) surface was made up of four atomic layers and oriented so that its normal vector corresponded to the *z*-axis. It was then modelled as prescribed by the GolP force field [54,55]. All Au atoms, including virtual sites, were frozen: only the dipoles used to reproduce the image charge effect of the Au (111) surface were allowed to reorient freely [55]. The capping layer was constructed by placing the ligands to reproduce a highly ordered and compact (√3 × √3)*R*30° overlayer [56]. The ligands’ parameters were generated with LigParGen, except for the S–Au, S–C, and Au–S–C bonded interactions, which were adapted from Ref. [57].

Periodic boundary conditions along each Cartesian coordinate were applied [58,59]. Both for the Au cluster and surface, the simulations were carried out in the canonical NVT ensemble (the amount of substance, N, the volume, V, and the temperature, T, are kept constant) with the simulation box conveniently scaled to recover the correct value of water density for the simple point charge (SPC) model [60]. The velocity-rescale algorithm developed by Bussi, Parrinello, and Donadio [61] was applied to keep the temperature constant at 25 °C (298.15 K). Simulations of H_2_TPPS^2−^ alone were instead performed in an isothermal-isobaric NPT ensemble (the amount of substance, N, the pressure, P, and the temperature, T, are kept constant). In this case, the Parrinello–Rahman barostat [62] was employed to ensure pressure coupling (1 bar).

Plain 1.0 nm cut-offs were applied for van der Waals and short-range electrostatic interactions. Long-range electrostatics were treated by means of the particle mesh Ewald method (PME) [63]. When dealing with the Au (111) surface, the PME method was combined with the correction proposed by Yeh and Berkowitz [64], which gives more accurate results for systems with slab geometry. To improve efficiency and reduce interactions with the periodic images in the *z*-direction, the simulation box was made at least three-times thicker than the real system [64], which was eventually sandwiched between two vacuum layers of the same size. For simulations involving only one functionalized surface, a second, plain Au slab was used to delimit the system on the upper face to avoid a surface–vacuum interface: this arrangement is also consistent with the one considered by Yeh and Berkowitz (water between two Pt walls) [64].

The simulation boxes ranged from 156 nm^3^ (two porphyrins without nanostructures) to 769 nm^3^ (‘hydrophobic’ dimer with Au_144_L_60_), depending on the system’s size: further details are provided in the Appendix A (Section 4). All starting configurations were properly minimized with a steepest-descent algorithm [58,59]. Initial velocities were assigned according to a Maxwell–Boltzmann distribution at 10 K, and a NVT equilibration followed. Classical equations of motion were integrated using a leapfrog algorithm and a time step of 0.5 fs [59,65]. Simulations of H_2_TPPS^2−^ alone also included a second equilibration step, under NPT conditions. For all systems, the time step was increased to 2 fs in the production run, and bonds to H atoms were constrained using the LINCS algorithm [66]. The results presented in the present article refer to straight 500 ns–1 μs MD simulations. A cluster analysis was performed with the gmx cluster tool by applying the gromos clustering method [67].

## 3. Results

### 3.1. Synthesis and Photophysical Characterization of the Components of the Hybrid Nanosystems

We decided to study the behaviour of the porphyrin in the presence of cationic gold NPs with different sizes and also to explore the effect of a different particle curvature. NPs with an average diameter of 11 ± 2 nm (BNPs) were prepared by a modified Turkevich procedure using citrate as reducing and surface stabilizing agent [47], while NPs with an average diameter of 2.6 ± 0.5 nm (SNPs) were prepared with a modified Brust and Scriffin protocol using dioctylamine as surface stabilizing agent and NaBH_4_ as reductant [49]. Both BNPs and SNPs were coated with a TMAO-SH shell (Figure 1b) by ligand exchange (for further information about the synthesis and characterization of the NPs, see Appendix A). The extinction spectra of the suspensions of the two NPs samples are reported in Figure 1c. As expected, BNPs featured an LSPR band at 520 nm, typical of gold nanoparticles with a core diameter larger than 3 nm. This resonance band is barely detected in SNPs, as expected for nanoparticles of this size. Variation of the pH in the interval 2–11 did not affect the NPs’ spectra. The thermogravimetric analysis allowed us to estimate similar values of surface ligand densities, with each thiol occupying 0.10 ± 0.02 nm^2^ of the BNPs’ surface and (0.09 ± 0.02) nm^2^ of the SNPs’ surface.

A Dynamic Light Scattering (DLS) analysis of BNPs at pH = 2 (Appendix A) yielded the average hydrodynamic diameter of 12 ± 3 nm, which might suggest a TMAO-SH shell of about 1 nm, as confirmed by TEM measurements previously reported [24]. Contrarywise, when the pH was set to 11, the hydrodynamic diameter increased to 28 ± 8 nm, suggesting the presence of a modest particles’ aggregation, not detectable in the extinction spectra. This phenomenon can be attributed to the increased concentration of OH^−^ ions, which likely bind to the nanoparticles’ surface more strongly than Cl^−^ ions and reduce the repulsion among the nanoparticles, favoring their aggregation.

Figure 1d reports the reference spectra of the porphyrin at 1 μM concentration. At a basic pH, the typical TPPS^4−^ spectrum, with the B band at 413 nm and the four Q bands at 515 nm, 550 nm, 580 nm, and 635 nm, was retrieved [1,16,68]. By decreasing the pH to 2.2, the spectrum turned into that of the protonated H_2_TPPS^2−^, with the B band shifted to 434 nm and the Q bands degenerated into two bands at 595 nm and 645 nm. Moreover, only a modest amount of J-aggregate was formed in the sample, as revealed by the appearance of the associated B bands at 490 nm and Q band at 710 nm [1,16,68]. Hence, in these conditions, H_2_TPPS^2−^ monomers are the dominant form.

Based on these results and the available literature, we pinpointed two main parameters expected to have the most relevant effect on the porphyrin aggregation in the presence of nanoparticles. The first was the pH, which was set at 2.2 and 11.0 to ensure that, in these conditions, only H_2_TPPS^2−^ and TPPS^4−^ were, respectively, present. The second parameter was the Particle Area Per Porphyrin (PAP), which expresses the dye concentration in terms of the ratio between the number of porphyrin molecules in the sample and the average nanoparticle surface area. In particular, we chose PAP values capable of ensuring either the complete saturation of the particles’ surface or a partial coating. To do so, we estimated that the nanoparticle surface coated by a single porphyrin should range from 0.4 nm^2^ (‘perpendicular’ orientation) to 2 nm^2^ (‘parallel’ orientation) of the NPs surface (schematic representation in Figure 1e) [69]. Consequently, we set the PAP values used in the experiments to 0.35 nm^2^ (surface saturation condition) and 3.5 nm^2^ (surface sub-saturation condition).

We performed a systematic spectroscopic investigation of the photophysical behaviour of the porphyrin in the presence of either SNPs or BNPs at the selected pH and PAP values. Each sample was independently prepared by adding the dye to a suspension of NPs adjusted at the required pH. Extinction and emission spectra were recorded after an eight-hour incubation at room temperature. The samples were then analyzed without any further washing/purification in order to preserve the PAP and the consequent sub-saturation/saturation of the NPs surface.

### 3.2. Porphyrin Aggregation in the Presence of BNPs

The first set of experiments was performed using BNPs at PAP = 0.35 nm^2^ (surface saturation conditions). The extinction spectra of the samples at pH 2.2 and 11 are reported in Figure 2a.

The analysis of the spectra at pH 2.2 confirmed the presence of the plexcitonic resonances. Indeed, the B band region was dominated by a pronounced dip at 490 nm, accompanied by two side peaks at about 477 and 540 nm. These features are clear evidence of the anticrossing behaviour typical of plexcitonic resonances. They formed upon mixing the LSPR of NPs and the 490 nm exciton resonance of H_2_TPPS^2−^ J-aggregates, as previously reported [30,70,71]. We labelled this plexciton resonance as ‘B plexciton’. Notably, the presence of this feature indicated the prevalent formation of nanoparticle-adsorbed H_2_TPPS^2−^ J-aggregates, as also confirmed by the diagnostic J-aggregate Q band present as a shoulder at 710 nm.

Another set of plexcitonic resonances appeared in the Q-bands region. Here, the typical dip was visible at about 650 nm, with a side peak at 670 nm. A second side peak at 620 nm was hidden by the nanoparticles plasmon band and is better visible at different PAP values (Figure 2b). These features revealed the formation of a ‘Q plexciton’, arising from the coupling between the strongest Q band of the monomeric H_2_TPPS^2−^ (645 nm) and the LSPR of aggregated NPs. The aggregation of BNPs in the presence of H_2_TPPS^2−^, which induced a red-shift of the LSPR band and reduced the detuning with the Q bands, was confirmed by DLS measurements (Appendix A) as well as by the TEM measurements previously reported [24].

Fluorescence measurements confirmed the behaviour previously recorded for similar systems. Indeed, we measured the typical emission of H_2_TPPS^2−^ monomers upon excitation at 430 nm and the weak emission of the Q plexciton upon excitation of the B plexciton at 510 nm (Figure 2d). This intriguing phenomenon was previously attributed to an inter-plexciton relaxation process [24].

Hence, in these conditions, BNPs template the self-assembly of H_2_TPPS^2−^ to form J-aggregates, which were barely present at the same pH and porphyrin concentration in the absence of nanoparticles. These aggregates, assembled on the particles’ surface, strongly coupled with the plasmon resonance to form plexciton B. However, the residual presence of H_2_TPPS^2−^ monomers not involved in aggregates has also been ascertained. At least in part, these monomers were bound to the BNP, as demonstrated by the formation of the plexciton Q.

It is relevant to note that the building up of an effective plexcitonic coupling requires that the transition dipole moments of the surface plasmons in the nanoparticle and the dyes are parallel [72]. For spherical NPs, the largest coupling is obtained when the transition dipoles of the dyes are orthogonal to the particle surface [73,74]. In our case, the porphyrin had two degenerate and perpendicular transition dipole moments, both laying on the molecular plane [1]. Therefore, it could be reasonably supposed that plexcitonic coupling in this system could be effectively established when the porphyrin molecular plane was perpendicular (or slightly tilted) to the NP’s surface. As we will discuss later, while such an arrangement is quite expected in the case of J-aggregates, it is not obvious in the case of monomers.

When the pH was raised to 11.0, the extinction spectrum of the hybrid sample (Figure 2a) appeared as the simple superimposition of those of NPs and TPPS^4−^. Furthermore, the fluorescence spectrum recorded at various excitation wavelengths always revealed only the typical emission of monomeric TPPS^4−^ (Figure 2d). Hence, the sample appeared to be composed mostly of small NPs aggregates (as proved by the moderate red shift of the LSPR band from 520 nm to 535 nm and confirmed by DLS investigations, Appendix A) and dye monomers. However, a closer inspection of the spectrum in the B bands region (Figure 2c) revealed that the absorbance around 400 nm was higher than expected if only monomers were present. This suggests that in this spectral region, there might have been contribution from another species, even if present in a small amount. B band signals in this position are diagnostic of the formation of H-aggregates, likely templated by the nanoparticles [17]. No plexcitons were observed at this pH. There are several possible reasons for this result. First, absorption bands of TPPS^4−^ and H-aggregates poorly overlaid with the LSPR of the nanoparticles. Second, it is quite obvious to expect the TPPS^4−^ molecules to be bound in parallel orientation onto the BNPs’ surface to optimize the ion pairing interaction with the cationic surface. However, this orientation would minimize the alignment of the transition dipole moments and reduce the strength of plexcitonic hybridization. Third, nanoparticle aggregation was much less relevant than at pH 2.2 (Appendix A), and this reduced the formation of nanogaps.

In the samples with PAP = 3.5 nm^2^ (partial surface coverage, Figure 2b), a different behaviour was observed. At an acidic pH, plexciton B was no longer visible in the extinction spectra, while plexciton Q could still be detected. Hence, in these conditions, the porphyrins concentration was so low that even the presence of the nanoparticles did not induce the H_2_TPPS^2−^ aggregation. Consequently, H_2_TPPS^2−^ monomers were the main species present in the sample and could build plexcitons with the NPs through nanogaps formation. No fluorescence could be detected for this sample upon excitation in the B plexciton region (510 nm). Clearly, the absence of the plexciton B hampered the inter-plexciton relaxation phenomenon observed in saturation conditions.

In the extinction spectra at pH = 11 (Figure 2b), a peak at 403 nm was the only visible feature besides a red-shifted BNPs plasmon band, suggesting the aggregation of the nanoparticles. As mentioned earlier, the band at 403 nm was attributed to the formation of H-aggregates. Such an attribution is supported by the fluorescence spectra where two peaks at 665 nm and 725 nm were recorded, in agreement with previous literature on H-aggregates (Figure 2d) [17].

At a basic pH, the particles templated the formation of H-aggregates in all the conditions investigated. In the subsaturation regime (PAP = 3.5 nm^2^), these were the only species present. This is somewhat counterintuitive because tetra-anionic TPPS^4−^ molecules present in the sample were not enough to saturate the NP surface, and consequently, one would expect TPPS^4−^ to bind to the cationic particle surface preferentially in the monomeric form. Interestingly, when the amount of TPPS^4−^ was increased to reach the saturation conditions (PAP = 0.35 nm^2^), H-aggregates became a minor species with respect to the large amount of TPPS^4−^ monomers free in solution [75]. The only possible explanation of this behaviour is that only a small fraction of the particle surface was available for porphyrin binding. Consequently, it was easily saturated by H-aggregates, even at subsaturation conditions, and when the porphyrin concentration was increased, most of the dyes could only remain free in the solution as monomers.

At an acidic pH, on the other hand, the system evolved more intuitively. At subsaturation conditions, only monomers were bound to the nanoparticles, likely in a perpendicular conformation that induced nanoparticles aggregation and allowed for plexciton formation. At a higher porphyrin concentration, nanoparticle-bound J-aggregates formed on the surface of the nanoparticles.

### 3.3. Characterization of the SNPs Nanosystems

Figure 3 reports the extinction spectra obtained by mixing SNPs with the porphyrin. In this case, being the SNP almost devoid of the LSPR band, observation of plexciton resonances was unlikely in any condition. In addition, the region in which the electromagnetic field was enhanced by SNPs (i.e., the effective volume [38]) could not be sufficiently large to include enough dye molecules to result in an observable plexciton [76,77].

In surface saturation conditions (PAP = 0.35 nm^2^, Figure 3a,b), as in the case of BNPs, we observed an extensive porphyrin J or H aggregation.

At an acidic pH, extinction spectra indicated that J-aggregates (peaks at 490 nm and 705 nm) were predominant. Fluorescence spectra confirmed their presence but revealed the additional presence of residual H_2_TPPS^2−^ monomers (Figure 4): upon excitation at 418 nm, we observed the typical emission of H_2_TPPS^2−^ monomers and a shoulder at 710 nm, characteristic of the J-aggregates [15,17], while excitation at 434 nm produced only the monomeric H_2_TPPS^2−^ emission spectrum.

At a basic pH (Figure 3a,b), the extinction spectra revealed instead the presence of H-aggregates, indicated by the band at 403 nm, and TPPS^4−^ monomers (band at 419 nm).

Surprisingly enough, in subsaturation conditions (PAP = 3.5 nm^2^, Figure 3c,d), the dye was always present as TPPS^4−^, irrespective of the pH. At an acidic pH (pH 2.2), only nanoparticle-bound TPPS^4−^ monomers were observed. Effective binding of TPPS^4−^ to the nanoparticles was confirmed by the fact that the samples were not fluorescent (data not shown), because of the quenching properties of the NPs [78]. Moreover, it must be noted that the position of the B band was red-shifted by 5 nm (from 414 nm to 419 nm). This suggests a certain degree of interaction between the conjugated aromatic rings of the porphyrin and the surface of the nanoparticle through phenomena like the image charge effect [79,80] or electromagnetic interactions [81].

In the sample at a basic pH, on the other hand, a shoulder at about 400 nm was visible in the spectrum and could be ascribed to the presence of H-aggregates beside the monomers (Figure 3c,d).

Hence, similarly to BNPs, SNPs also template the formation of H-aggregates at basic pH and J-aggregates at an acid pH. At a basic pH, the increase of the dye amount resulted in the rise of the fraction of H-aggregates over the particle-bound monomers. It is, however, noteworthy that, also in this case, H-aggregates were observed in subsaturation conditions. At an acidic pH, the system changed from nanoparticle-bound monomers, which, in this case, were deprotonated TPPS^4−^ molecules, to nanoparticle-bound J-aggregates. Apparently, the dyes aggregation is accompanied at this pH by their protonation. Indeed, the TPPS^4−^ form, detected as the only species present in the samples at a high PAP, did not form J-aggregates.

### 3.4. MD Simulations

Classical atomistic MD simulations were also performed to gain further insights into the supramolecular arrangements of porphyrins on the NPs. We focused our attention on the species at an acidic pH, since plexciton formation, as well as the less intuitive conformations, were observed in these conditions. Various representative model systems were designed to capture (and possibly emphasize) the essential features of the different configurations while reducing the overall complexity of the examined assemblies.

BNPs were likely seen by an approaching molecule or dimer as a locally flat surface. Therefore, an Au (111) surface (the most stable and, typically, the most abundant in spherical nanoparticles) [82,83], fully coated with –S-AO ligands (–S(CH_2_)_8_NH_3_^+^, ligand density = 4.48 ligands/nm^2^), was chosen to model BNPs [84]. The use of ammonium rather than trimethylammonium heads (as in the experiments) simplified the computational setup, keeping the main electrostatic feature of the charged heads.

A one-μs MD simulation was initially run for a single porphyrin molecule laid above the ligands, with the molecular plane in a ‘parallel’ arrangement with respect to the surface (Appendix A). The trajectory was first analyzed by considering the H_2_TPPS^2−^–ligand minimum distance. This distance distribution was mainly concentrated (~52% of the overall trajectory) below 0.30 nm (Appendix A), a value consistent with a direct porphyrin–capping layer interaction. The majority of the small distance frames showed only one SO_3_^−^ group in close contact with the ligand, with the remaining portion of the molecule stretching into the solution (Figure 5a). Such an arrangement was likely due to a compromise between the favorable SO_3_^−^–NH_3_^+^ electrostatic interaction and the unfavorable electrostatic repulsion that the positive porphyrin’s core would experience with the capping layer by staying flat on it. The concurrent presence of many Cl^−^ counterions, which were mainly concentrated around the charged headgroups of the ligands, likely played a competitive role with SO_3_^−^ binding, possibly explaining why binding via only one SO_3_^−^ (instead of two) was often observed.

The most remarkable feature was indeed the angle, α, between the normal vector of the porphyrin plane and that of the surface plane (the latter being coincident with the *z*-axis), which provides information of the relative orientation of the molecular plane with respect to the surface (0° is the parallel orientation, 90° is the perpendicular). Its average value was 68.9°, while the associated probability continually increased up to a maximum of 87.4° (Appendix A). This evidence clearly suggests the existence of a preferred arrangement, where the H_2_TPPS^2−^ plane was almost perpendicular to the underlying ligand layer.

The previous findings were substantially confirmed by a 600-ns simulation for a porphyrin molecule already placed perpendicularly to the capping-layer and with a bidentate coordination initially enforced (Appendix A).

An H_2_TPPS^2−^ dimer was chosen as the prototypical example of a porphyrins’ J-aggregate; a preliminary simulation was therefore carried out to provide a reasonable starting structure of the dimer itself. As detailed in the Appendix A, we observed two possible dimer conformations; the one that was more persistent in the simulations was kept for the simulations with the NPs. A one-μs-long MD simulation was also performed for such a dimer on the same model of the NPs facets used here above for the monomer. During the entire trajectory, the dimer kept its internal structure unchanged. Notably, the number of frames where H_2_TPPS^2−^ was at an interaction distance with the ligands summed up to 85% of the total, with a substantial increase from the single porphyrin’s case (~52%, see above). This evidence points to a strongly enhanced interaction between the dimer and the capping layer compared to the monomer.

A cluster analysis highlighted the preference for a conformation with four effective (two per each porphyrin) SO_3_^−^–NH_3_^+^ ion pairs: more than four out of five analyzed frames (~83%) could be traced back to the same kind of arrangement (Figure 5b). Hence, besides resulting in a stronger interaction with the NP, aggregation greatly favored an almost perpendicular orientation with respect to the surface.

The results reported so far show that H_2_TPPS^2−^ preferentially adopted a perpendicular conformation on BNPs suitable to form plexcitons and to act as a cross-linker with other NPs. To further investigate the cross-linking ability of H_2_TPPS^2−^, we performed a one-μs MD simulation of a single porphyrin inside a box enclosed by two functionalized Au (111) surfaces. As before, the porphyrin was initially laid above one set of ligands, while the surfaces were fixed at a distance that allowed for introducing a solution layer 1.8-nm-thick between the facing capping layers (Appendix A). The frames with at least one porphyrin–ligands contact were selected for the cluster analysis: they represented ~90% of the entire trajectory.

A bridging geometry was clearly adopted in ~8% of the analyzed snapshots (Figure 6a), but an additional 35% of frames were characterized by a contact distance between one of the porphyrin’s SO_3_^−^ group and the second capping layer only slightly above (0.35–0.40 nm) the threshold value that we chose to identify contact (0.30 nm, see Appendix A). Overall, the H_2_TPPS^2−^ was found in a position compatible with a cross-linking role between the two surfaces for almost half of the simulation time. In a second shorter (500 ns) simulation, after the distance between the two opposite capping layers was reduced by 0.2 nm, the percentage of structures corresponding to a cross-linking arrangement of the dye molecule rose to ~80% (Figure 6b). Although some structures were able to exploit all the four possible binding sites, the H_2_TPPS^2−^ preferably interacted with the two capping layers only via two opposite SO_3_^−^ groups, as in Figure 6b. The latter arrangement was clearly favored because of the competitive association of chloride counterions with the cationic headgroups on the ligands. Due to the larger dimensions of H_2_TPPS^2−^ in that direction, the molecule was then forced to a more tilted orientation with respect to the normal vector of the two surfaces (i.e., to the *Z*-axis). This was reflected by the average angle between the porphyrin plane and the *Z*-axis (60.7°and 47.3° for the thicker and the thinner solution layer, respectively; see Appendix A), while that obtained for a tetra-coordinated structure approached the ideal value of 90.0° for a perfectly “standing” porphyrin.

Analogous results were retrieved after the same simulations (i.e., 1 μs with a larger box, 500 ns with a thinner box) were carried out for an H_2_TPPS^2−^ molecule initially placed in a standing position between the two capped surfaces (see Appendix A).

In order to investigate the NP curvature effects, a cluster composed of 144 gold atoms, Au_144_ (S-AO)_60_, was chosen to mimic small NPs. The cluster core had a diameter around 1.8 nm, close enough to that of the SNP used in the experiments, and had already been reported to well describe their behavior [85,86,87,88]. This time, both the porphyrin and the dimer were initially separated by more than 2 nm from the nanostructure and allowed to autonomously find their way to the NP. A one-μs-long MD simulation was carried out with a non-neutralized box to avoid kinetic barriers due to the competition of Cl^−^ counterions for the binding upon NH_3_^+^ heads. The results, presented in the following, were then confirmed by 500 ns of an additional simulation, including enough Cl^−^ counterions to neutralize the box (Appendix A).

In these simulations, the formation of the first nanocluster-H_2_TPPS^2−^ assembly took just 10 ns: in the resulting complex, the porphyrin was perpendicularly oriented, and half penetrated in the capping layer (Figure 7a). However, a stable arrangement was reached only after 100 ns: the H_2_TPPS^2−^ molecule progressively changed its orientation from radial to tangential with respect to the gold cluster, and this movement came with a reorganization of the underlying ligands to form a bowl-like cavity, which the porphyrin fitted into (Figure 7b,c) [85,86]. Such a conformation maximized the SO_3_^−^–NH_3_^+^ contacts while preventing the repulsion between the same NH_3_^+^ headgroups and the protonated pyrroles on the H_2_TPPS^2−^ core. Notably, this conformation is also compatible with a favorable hydrophobic interaction between the phenyl groups of the porphyrin and the alkyl chains of the opened-up capping layer. Once reached, this conformation remained stable until the end of the one-μs trajectory, with only minor adjustments of the phenyl substituents, as confirmed by the plot of the distance between the centers of mass (c.o.m.) of the porphyrin and of Au_144_ (Appendix A).

A one-μs-long MD simulation was finally carried out combining the same Au_144_L_60_ cluster with the dimer. Without counterions, the association again occurred within just 10 ns, with both porphyrins setting effective SO_3_^−^–NH_3_^+^ interactions with the ligands and penetrating for a few Å in the capping layer. Only minor changes were observed throughout the rest of the simulation (Figure 8): the most important one occurred around 730 ns, when one of the H_2_TPPS^2−^ residues moved deeper into the capping layer, leading to slightly different values of the c.o.m.–c.o.m. distances between Au_144_ and the two porphyrins (1.83 nm and 2.04 nm, Appendix A). Looking only at the dimer, the association with the nanocluster did not significantly affect its internal conformation, as proven by the c.o.m.–c.o.m. separation between the molecules and by the slightly larger angle between their planes (0.78 nm and 15.3°, respectively). Again, all these features were retained in a subsequent 500 ns trajectory, stemming from the final conformation of the previous 1 μs MD, after box neutralization (Appendix A).

## 4. Discussion

Overall, the results reported here reveal that the interaction of cationic gold nanoparticles with the tetra−4-sulfonyl-phenyl-porphyrin TPPS produces a series of different structures depending on the dye/nanoparticle ratio, the pH, and the particle size. These conformations are graphically summarized in Figure 9.

At first, our results confirm the early observation that, when the dye is present in a sufficient amount to saturate their surface, the nanoparticles induce J-aggregation of H_2_TPPS^2−^ and H-aggregation of TPPS^4−^ [14,21,22,23].

What is surprising is the fact that H-aggregates are formed at a basic pH, even in subsaturation conditions, both with large and small NPs. TPPS^4−^, which is the species present in these conditions, has four negative charges on the peripheral sulfonate groups. These groups generate a relevant charge repulsion that counterbalances the hydrophobic effect and prevents H-aggregation. However, the strong negative charge, while preventing aggregation of the dyes in solution, favors their absorption on the cationic NPs’ surface. Indeed, the binding of a tetra-anionic molecule to a cationic ligand shell-protected gold NP was reported to be very strong [48,89,90]. Simple considerations based on the Coulomb law suggest that the optimal interaction is reached when the molecule lies flat on the particle surface (Appendix A).

In this position, the distance between the sulfonate groups and the ammonium headgroups of the coating thiols is minimized. This parallel orientation likely promotes the formation of H-aggregates, with additional dye molecules stacking on the adsorbed one, whose charge is neutralized, at least partially, by the underlying cationic monolayer. At first sight, this process should occur only in saturation conditions, since, in subsaturation, the dye molecules should prefer surface binding to stacking. On the contrary, we observed H-aggregation in subsaturation conditions and the presence of a large excess of non-aggregated dyes in saturation ones. We speculate that this counterintuitive behaviour could be ascribed to the presence of OH^−^ ions forming close ion pairs with the ammonium headgroups. If this were the case, OH^−^ ions would reduce both the electrostatic stabilization of the NPs and the NP surface available for ion pairing interactions. Noticeably, the expected reduction of colloidal stability was confirmed by the DLS experiments discussed above, which indicated that BNPs undergo some aggregation at pH 11. The reduction of the surface available is the most likely explanation for the large excess of free dyes in saturations conditions, as the amount of porphyrin present largely exceeded the binding sites available on the particles surface, as well as the observation of H-aggregates also in subsaturation conditions. Indeed, the decrease of the dye amount was, in this case, counterbalanced by the small number of binding sites on the particles. At the same PAP of 3.5 nm^2^, the formation of H-aggregates was quantitative in the case of BNPs and only partial with SNPs. This suggests a greater surface area reduction for BNP than for SNP, possibly because the binding of OH^−^ ions could be less effective in the case of smaller NPs. Interestingly, also MD simulations, even if performed on slightly different systems, suggest a relevant role of counterions in modulating the interactions of poly-charged molecules with the NP’s surface.

At an acidic pH, other unexpected behaviors were observed. When TPPS^4−^ was converted in the H_2_TPPS^2−^ form, the affinity for the NPs’ surface was reduced but remained significant. Simple electrostatic considerations, based on rigid charge distribution models, suggest that also in this case the most stable orientation should be the parallel one. In the perpendicular position, only one or two sulfonate groups are in contact with the surface, while the others are more or equally distant from the surface than the positively charged pyrroles. Since the Coulomb repulsion experienced by the pyrroles is stronger than the attraction experienced by the outer sulfonate groups, the perpendicular configuration should be less stable. However, MD simulations showed that such a picture is oversimplified. Indeed, the parallel configuration is preferred only if a massive reorganization of the monolayer occurs to decrease the resulting unfavorable interactions (charge repulsion, poor solvation of the cationic headgroups), as in the case of SNP. When the monolayer rigidity is increased because of the decreased curvature [91], as in the case of BNP, such reorganization is impossible, and the porphyrin prefers a perpendicular arrangement.

Of course, MD simulations do not account for the possibility that the molecule undergoes deprotonation. Experimental results suggest that the porphyrin bound to SNP in the parallel conformation is still an unstable state that further stabilizes by deprotonating the pyrrole nitrogens. This effect is similar to that described with cationic maghemite nanoparticles of a similar size [23]. The fact that deprotonation occurs with small NPs and not with the large ones suggests that this is possible only after the parallel conformation is reached. Moreover, MD simulations show, for such parallel conformation, the proximity of the pyrrole nitrogens to the hydrophobic alkyl chains of the ligands, which certainly disfavors their charged (i.e., protonated) over neutral (i.e., deprotonated) form.

The most remarkable result of this study is hence the evidence that different binding modes and even chemical features of the same molecule are enabled by different particle curvatures, which, in turn, strongly affect the mobility of the coating molecules. In particular, the perpendicular conformation of H_2_TPPS^2−^, which is ideal both for the coupling with the nanoparticle plasmon to form the plexciton and for the crosslinking of the particles to form the hotspot needed to enhance the coupling, is forced by the rigidity of the monolayer coating BNPs.

MD simulations confirmed that, regardless of the initial orientation, there is a marked tendency of H_2_TPPS^2−^ to form ion-pair interactions with ligands of two particles in a bridging arrangement between the two capping layers. This remains true even when the molecule, as for the larger box, possesses enough space to reorient freely or to lay preferentially closer to one surface [24].

The perpendicular orientation and the crosslinking ability are apparently maintained also when J-aggregates are considered, in agreement with the observation of strong spectroscopic signatures for the plexcitonic coupling of H_2_TPPS^2−^ J-aggregates and big NPs. MD simulations confirm this result, as well as the greater affinity of the J-aggregates for the particles compared to their monomeric form, which suggests that the binding of monomeric dyes to the surface of colloidal NPs is less effective than that of J-aggregates. The low binding constant of monomeric dyes was already proposed [44,92] as the reason for the less-marked anticrossing behaviour of the plexcitonic resonances in the extinction spectrum [24].

## 5. Conclusions

This study revealed that the interaction between TPPS and cationic nanoparticles can result in the formation of a large variety of different structures. Overall, the self-assembly behaviour of the system is controlled not only by the balance of electrostatic and hydrophobic interactions established by the two entities, but also by their structural rigidity. TPPS molecules, in all the protonation forms studied, are strongly attracted by the cationic NPs’ surface. Nanoparticle binding attenuates the electrostatic repulsion between the dye molecules favoring the formation of aggregates, whose structure is controlled by the charge distribution of the dye itself. It is noteworthy that the elusive TPPS^4−^ H-aggregates are easily formed both with big and small NPs, suggesting that this is a general process and should be observed with most cationic NPs. On the other hand, the formation of J-aggregates from H_2_TPPS^2−^ in the presence of polycationic species is a quite-recognized process, and the behaviour of cationic NPs is in line with expectations.

The most elusive entity observed in this system is the Q plexciton, which consists in the coupling of the LSPR of aggregated BNPs and H_2_TPPS^2−^. The literature reports only a few other examples of plexcitons formed with non-aggregated molecules and dispersed nanoparticles (see for instance ref. [41,42,43,44]) and none with porphyrin dyes. Furthermore, to our knowledge, this is the only example of a plexcitonic set of resonances that occurs in solution using an LSPR of a plasmon nanogap. Our results indicate that its formation is the result of a delicate balance of mutual interactions and structural factors. The high density of the BNP coating monolayer forces H_2_TPPS^2−^ into the perpendicular conformation, which is essential to allow the strong plasmon–exciton coupling. In this conformation, the rigid and multiply charged dye can also behave as an effective particle cross-linker [93], inducing NPs aggregation to form the nanogaps necessary to enhance its absorption and to shift the plasmons’ position to meet the coupling conditions.

Studies aimed to improve further our ability to control the self-organization of coupled plasmonic nanosystems are being performed in our labs.

## Figures and Tables

**Figure 1 nanomaterials-12-01180-f001:**
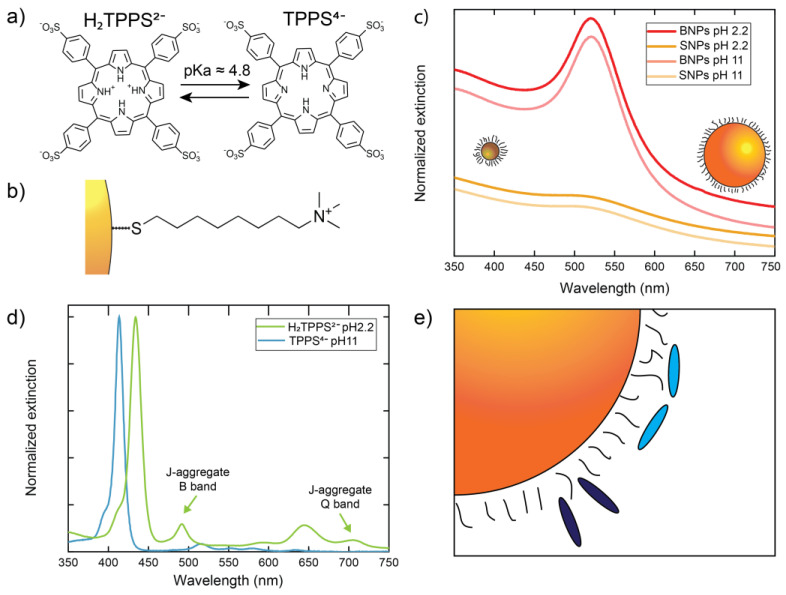
Components of the hybrid systems. (**a**) Molecular structures of diprotonated and free base porphyrin, labelled H_2_TPPS^2−^, and TPPS^4−^, respectively. (**b**) Representation of the TMAO-SH capping layer molecule. (**c**) Extinction spectra of big (BNPs) and small (SNPs) nanoparticles at different pH. The spectra are shifted vertically to ease the comparison. The cartoons of big and small nanoparticles are represented in scale. (**d**) Normalized extinction spectra of H_2_TPPS^2−^ at pH = 2.2 and of TPPS^4−^ at pH = 11 (1 μM solutions in MilliQ water). The two arrows indicate the B and Q bands of TPPS J-aggregate, allowing them to be distinguished from the bands of the monomer. (**e**) Schematic representation of the perpendicular (dark blue) and parallel (light blue) orientation of the porphyrins on a nanoparticle surface.

**Figure 2 nanomaterials-12-01180-f002:**
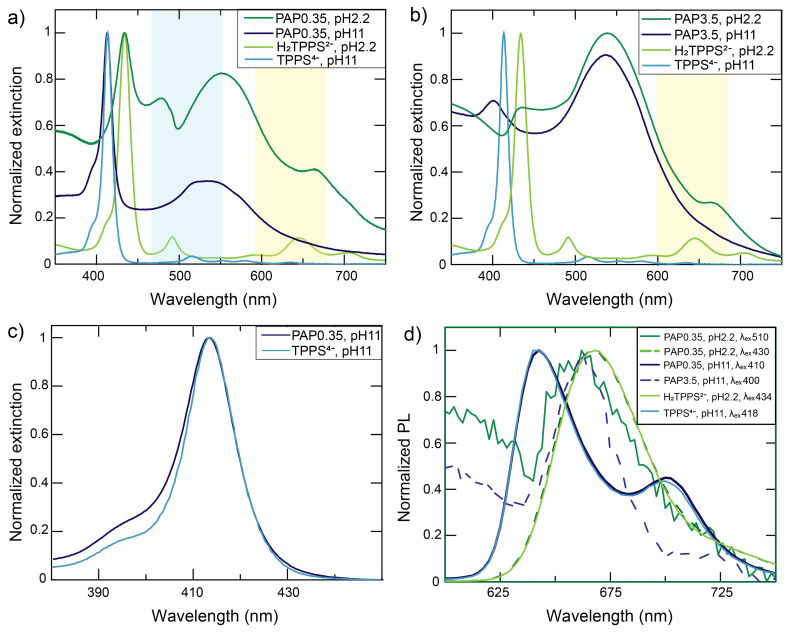
Overall perspective of the extinction and emission spectra of BNPs. (**a**) Extinction spectra of the hybrids at PAP = 0.35 nm^2^. H_2_TPPS^2−^ and TPPS^4−^ absorption spectra at pH 11 and 2.2, respectively, are reported as references. (**b**) Same as (**a**) but for PAP = 3.5 nm^2^. The blue (yellow) area highlights the presence of B (Q) plexciton resonances. (**c**) Comparison between the normalized extinction spectra of TPPS^4−^ and BNPs at PAP = 0.35 nm^2^ and at pH = 11. A constant baseline subtraction has been performed to ease the comparison. The TPPS^4−^ absorption spectrum is reported as a reference. (**d**) Normalized emission spectra of nanohybrids in different conditions compared with the emission spectra of H_2_TPPS^2−^ (pH = 2.2) and TPPS^4−^ (pH = 11). The excitation wavelengths are reported in nm.

**Figure 3 nanomaterials-12-01180-f003:**
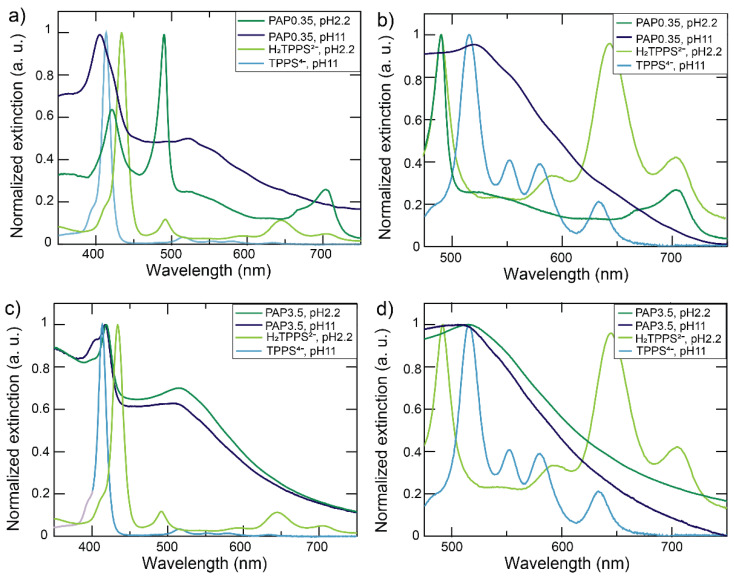
Overall perspective of the extinction spectra of SNPs. Normalized extinction spectra of SNPs, H_2_TPPS^2−^ (pH = 2.2), and TPPS^4−^ (pH = 11) at (**a**) PAP = 0.35 and (**c**) 3.5 nm^2^. A zoom of the 480–750 nm region of each plot is reported in panel (**b**) and (**d**), respectively.

**Figure 4 nanomaterials-12-01180-f004:**
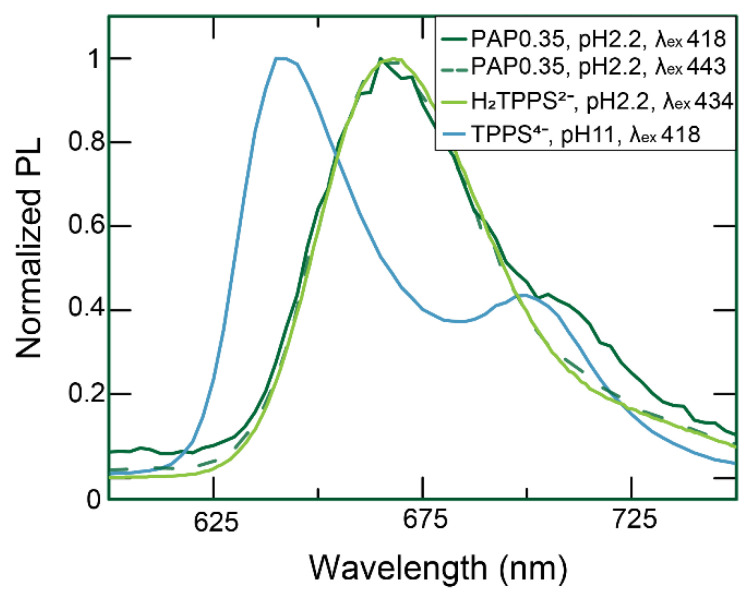
Overall perspective of the emission spectra of SNPs. The normalized emissive spectra of the nanohybrid sample prepared with SNPs at PAP = 0.35 nm^2^ were compared with the spectra of H_2_TPPS^2−^ (pH = 2.2) and TPPS^4−^ (pH = 11) as references.

**Figure 5 nanomaterials-12-01180-f005:**
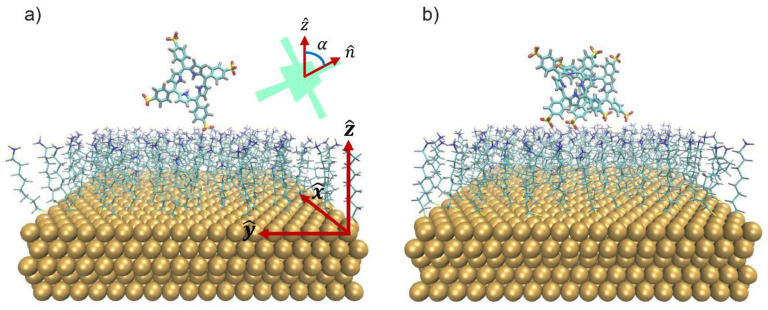
The central structures of the most representative clusters (33.2% and 82.7% of the analyzed frames, respectively). They were obtained from the one-μs simulations, starting from a ‘parallel’ porphyrin (**a**) and the ‘hydrophobic’ dimer (**b**). As in the following, water, and Cl^−^ counterions are not shown for clarity. The inset shows, schematically, how the angle, α, between the normal vector to the porphyrin plane and the *Z*-axis was defined.

**Figure 6 nanomaterials-12-01180-f006:**
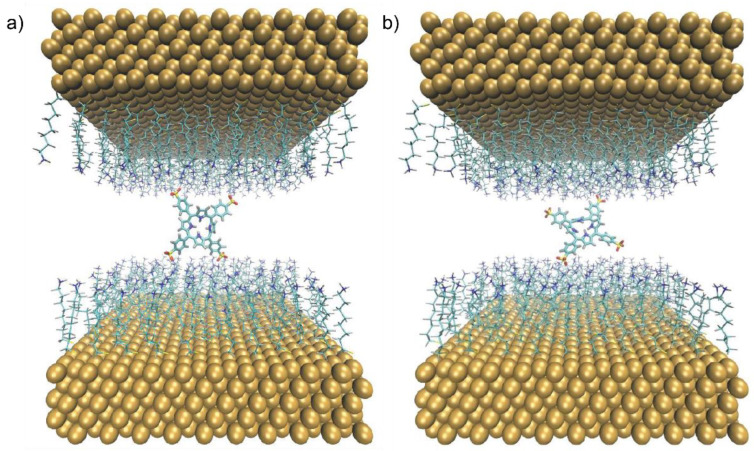
The central structures of two representative clusters, obtained from the simulations of a dye molecule initially laid above the capped surface at the bottom. The central structure in (**a**) was obtained from the 1 μs trajectory with the thicker box, while in (**b**), it was from the 500 ns trajectory, with a thinner solution layer. They account for 5.5% and 31.6% of the analyzed frames in the corresponding trajectories.

**Figure 7 nanomaterials-12-01180-f007:**
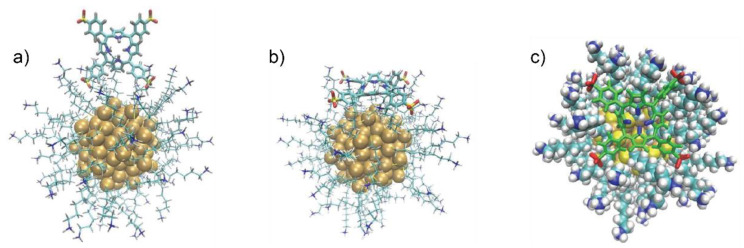
Significative representation of the nanocluster H_2_TPPS^2−^ supramolecular assembly. In (**a**), it is represented after 10 ns. (**b**) and (**c**) both depict the final conformation, after 1 μs MD. A different representation is chosen in (**c**) to emphasize the formation of a bowl-like cavity in the ligand layer; SO_3_^−^ peripheral groups and the pyrrolic protons are highlighted in red and blue, respectively.

**Figure 8 nanomaterials-12-01180-f008:**
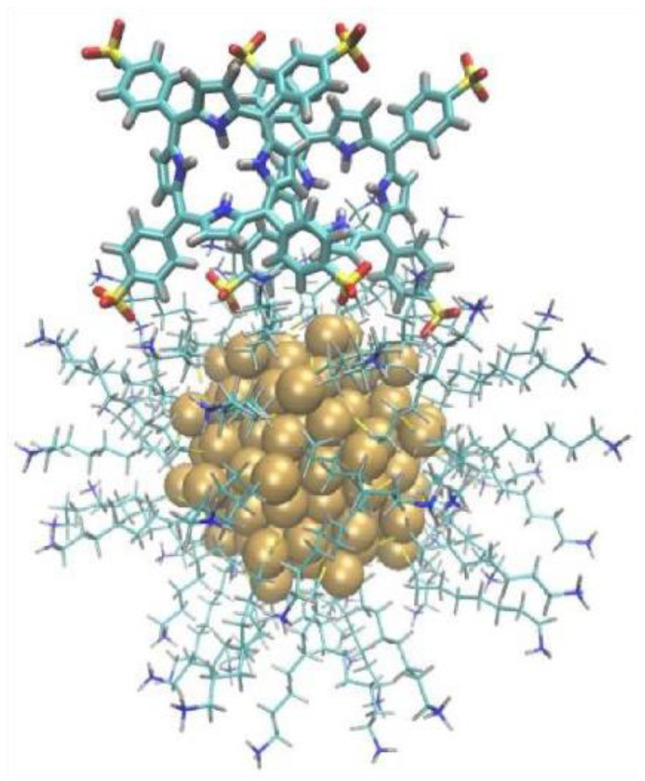
The nanocluster-dimer supramolecular assembly after 1 μs MD. The two porphyrin residues are both radially oriented with respect to the Au_144_L_60_ cluster.

**Figure 9 nanomaterials-12-01180-f009:**
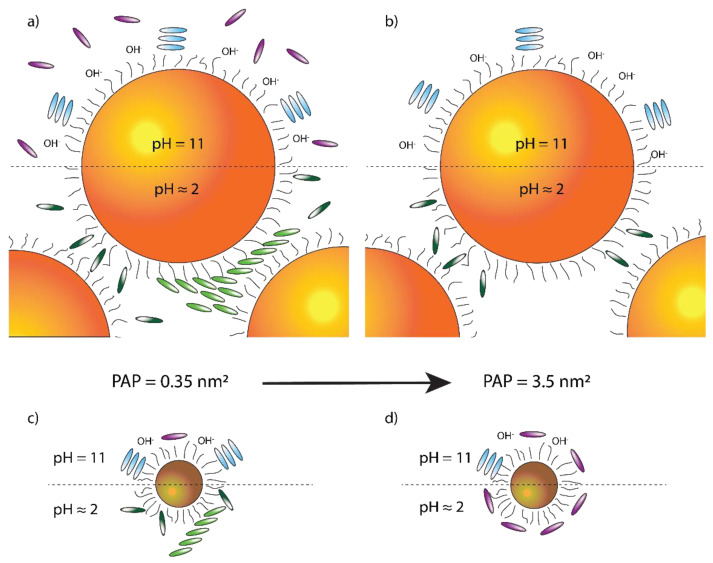
Overview of the interactions between NPs and porphyrin dyes. The structures templated by BNPs and SNPs are schematized in panels (**a**), (**b**), (**c**), (**d**), respectively. Interactions at pH = 11 (pH = 2.2) are at the top (bottom) of each image. Panels (**a**) and (**c**) refer to PAP = 0.35 nm^2^, while panels (**b**) and (**d**) to PAP = 3.5 nm^2^. Each porphyrin species is pinpointed by a different color: monomeric TPPS^4−^ (H_2_TPPS^2−^) is violet (dark green), while H (J) aggregates are light blue (light green).

**Table 1 nanomaterials-12-01180-t001:** Volumes used for the preparation of the nanohybrids samples.

Title 1	HCl/NaOH, μL	AuNPs (5 mM), μL	TPPS (1 mM), μL
BNPs, PAP = 0.35	998	1	1
BNPs, PAP = 3.5	994	6	0.5
SNPs, PAP = 0.35	998	0.5	1
SNPs, PAP = 3.5	996	2.5	1

## Data Availability

The data presented in this study are reported in the article and in the Supplemetary Material. Additional details can be requested to the corresponding authors.

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
