# Peer review of "Engineering the Aggregation of Dyes on Ligand-Shell Protected Gold Nanoparticles to Promote Plexcitons Formation"

_nanomaterials, 2022, doi:10.3390/nano12071180_

Round 1

Reviewer 1 Report

The article is interesting and relevant research devoted to the formation of plexcitons in suspensions of hybrid nanosystems prepared by coupling cationic gold nanoparticles to tetraphenyl porphyrins. The authors studied in detail the effect of such parameters as nanoparticle size, pH of the solution, and concentration on the interaction of nanoparticles with porphyrins, and also carried out atomistic molecular dynamics simulations, complementative the obtained experimental results. The main results of the article were confirmed experimentally and supported by computational investigations. The results obtained in the work make a significant contribution to the study of plexcitonic systems and porphyrins aggregates and contribute to further progress in this area and the article can be accepted almost in its current form.

There is a comment related to the design of the article. In Figures 2 and 3, the dark blue and dark green colors are very difficult to distinguish, and it would be better to choose other colors to better visualize these figures.

Author Response

"The article is interesting and relevant research devoted to the formation of plexcitons in suspensions of hybrid nanosystems prepared by coupling cationic gold nanoparticles to tetraphenyl porphyrins. The authors studied in detail the effect of such parameters as nanoparticle size, pH of the solution, and concentration on the interaction of nanoparticles with porphyrins, and also carried out atomistic molecular dynamics simulations, complementative the obtained experimental results. The main results of the article were confirmed experimentally and supported by computational investigations. The results obtained in the work make a significant contribution to the study of plexcitonic systems and porphyrins aggregates and contribute to further progress in this area and the article can be accepted almost in its current form."

We thank Reviewer 1 for the positive comments

"There is a comment related to the design of the article. In Figures 2 and 3, the dark blue and dark green colors are very difficult to distinguish, and it would be better to choose other colors to better visualize these figures."

We provided a new version of figures 2 and 3 were colord and lines boldness were changed to improve visualization.

Reviewer 2 Report

The present paper is a continuation of  previous paper published by the same authors, i.e. N. Peruffo et al.,  Nanoscale 2021, 13, 6005-6015 (Ref. [24]). It was a good idea to continue research and investigate an influence of different parameters (size of nanoparticles, pH of solution, and concentration of porfirin molecules)  on the interaction between nanoparticles and porphyrins. Experimental studies are supplemented by molecular dynamics simulations. As a result of comprehensive investigations a set of various structures formed by porphyrin dyes have been proposed. This is an interesting and well-written paper which shows how the formation of plexcitons can be controlled. The light-matter interaction in nanosystems is an important and modern field of research. Moreover, the formation and properties of plexcitions is a hot topic.  In my opinion the paper should be accepted for publication in Nanomaterials. Nevertheless, I have some small remarks:  

  • Page 3, line 125. One can see two neighboring commas.
  • Page 5, lines 194-195. In units nm2 superscripts are necessary.
  • Page 6, line 256. We read  “…two side peaks at 620 nm and 670 nm.” However, in Fig. 2a we see only the peak at 670 nm. Most probably the band 620 nm is hidden in the right-side wing of the band centered at about 550 nm, nevertheless even the smallest “bump” is not seen.
  • Page 8, Fig. 2d. The lines PAP0.35, pH11, λex410 and TPSS4-, pH11, λex418 overlap so strongly that it is difficult to distinguish them. It would be better to use more contrasting colors.
  • Page 10, Fig. 3. There are two figures “c” but no figure “b”. Moreover, in right panels zooms of the 480-750 nm region are displayed and the ordinate axis “Normalized extinction” are shown with units, though for different plots units are different. It would be better to display axes without units and describe them as “Normalized extinction (arbitrary units)” or “Normalized extinction (a. u.)”
  • Page 11, Fig.5. In the figure caption one can read that “The inset shows schematically…”, however in the figure we see no insert.
  • Page 15, Fig. 9. In the figure caption we read “… monomeric TPPS4- (H2TPPS2-) is blue (dark green), while H(J) aggregated are light blue (light green).” However, in the figure we see the colors: wine (or purple) and light blue. Moreover, it is nearly impossible to distinguish between dark green and light green.      

Author Response

"The present paper is a continuation of  previous paper published by the same authors, i.e. N. Peruffo et al.,  Nanoscale 2021, 13, 6005-6015 (Ref. [24]). It was a good idea to continue research and investigate an influence of different parameters (size of nanoparticles, pH of solution, and concentration of porfirin molecules)  on the interaction between nanoparticles and porphyrins. Experimental studies are supplemented by molecular dynamics simulations. As a result of comprehensive investigations a set of various structures formed by porphyrin dyes have been proposed. This is an interesting and well-written paper which shows how the formation of plexcitons can be controlled. The light-matter interaction in nanosystems is an important and modern field of research. Moreover, the formation and properties of plexcitions is a hot topic.  In my opinion the paper should be accepted for publication in Nanomaterials."

We thank Reviewer 1 for his positive comments 

"Nevertheless, I have some small remarks:  

  • Page 3, line 125. One can see two neighboring commas."

Corrected.

  • Page 5, lines 194-195. In units nm2 superscripts are necessary.

Corrected, as well as other similar typos.

  • Page 6, line 256. We read  “…two side peaks at 620 nm and 670 nm.” However, in Fig. 2a we see only the peak at 670 nm. Most probably the band 620 nm is hidden in the right-side wing of the band centered at about 550 nm, nevertheless even the smallest “bump” is not seen.

Reviewer 2 is right, the band at 620 nm is not visibile in Figure 2a, while it is visible in figure 2b as shoulder as well as in spectra reported in ref. 24. We modified the text accordingly: "Here, the typical dip is visible at about 650 nm, with a side peak at 670 nm. A second side peak at 620 nm is hidden by the nanoparticles plasmon band and is better visible at different PAP values (Figure 2b)."

  • Page 8, Fig. 2d. The lines PAP0.35, pH11, λex410 and TPSS4-, pH11, λex418 overlap so strongly that it is difficult to distinguish them. It would be better to use more contrasting colors.

A new version of figure 2 was provided, as also suggested by Reviewer 1.

  • Page 10, Fig. 3. There are two figures “c” but no figure “b”. Moreover, in right panels zooms of the 480-750 nm region are displayed and the ordinate axis “Normalized extinction” are shown with units, though for different plots units are different. It would be better to display axes without units and describe them as “Normalized extinction (arbitrary units)” or “Normalized extinction (a. u.)”

A new version of figure 3 was provided, as also suggested by Reviewer 1, with errors corrected and the modification suggested made.

  • Page 11, Fig.5. In the figure caption one can read that “The inset shows schematically…”, however in the figure we see no insert.

We thank the reviewer for signaling this issue. A new version of figure 5 with the inset was provided.

  • Page 15, Fig. 9. In the figure caption we read “… monomeric TPPS4- (H2TPPS2-) is blue (dark green), while H(J) aggregated are light blue (light green).” However, in the figure we see the colors: wine (or purple) and light blue. Moreover, it is nearly impossible to distinguish between dark green and light green.   

A new version of figure 9 with color changed to improve visualization was provided.